# Fire as a Major Factor in Dynamics of Tree-Growth and Stable $\delta^{13}$C and $\delta^{18}$O Variations in Larch in the Permafrost Zone

Anastasia A. Knorre [1,2,*], Rolf T. W. Siegwolf [3], Alexander V. Kirdyanov [1,4], Matthias Saurer [3], Olga V. Churakova (Sidorova) [1,3] and Anatoly S. Prokushkin [1,4]

1 Institute of Ecology and Geography, Siberian Federal University, 660041 Krasnoyarsk, Russia; kirdyanov@ksc.krasn.ru (A.V.K.); ochurakova@sfu-kras.ru (O.V.C.); prokushkin@ksc.krasn.ru (A.S.P.)
2 Science Department, National Park "Krasnoyarsk Stolby", 660006 Krasnoyarsk, Russia
3 Ecosystem Ecology, Forest Dynamics, Swiss Federal Institute for Forest, Snow and Landscape Research WSL, 8903 Birmensdorf, Switzerland; rolf.siegwolf@wsl.ch (R.T.W.S.); matthias.saurer@wsl.ch (M.S.)
4 V.N. Sukachev Institute of Forest SB RAS, The Separate Department of Federal Research Centre KSC SB RAS, 660036 Krasnoyarsk, Russia
* Correspondence: aknorre@ksc.krasn.ru; Tel.: +7-(904)-896-03-30

**Abstract:** Wildfires are one of the most important environmental factors controlling forest ecosystem physiology and the carbon balance in the permafrost zone of North Siberia. We investigated tree-ring width (TRW) and stable isotope chronologies in tree-ring cellulose ($\delta^{13}$C$_{Cell}$, $\delta^{18}$O$_{Cell}$) of *Larix Gmelinii* (Rupr.) Rupr. from a wet (WS) and a dry (DS) site. These sites are characterized by different fire histories (fire in 1852 at the wet and 1896 at the dry sites, respectively). TRW and $\delta^{18}$OCell are identified to be the most sensitive parameters in the changing tree growth conditions after fire. The differences in the soil seasonal thermal regime of sites after fires are shown in the relationship between the studied parameters. The $\delta^{13}$C$_{Cell}$ values in tree rings from the two sites are positively correlated independently of the fire impact. This fact indicates that $\delta^{13}$C$_{Cell}$ chronologies might be more adequate for climatic reconstruction in the region due to the climate signal consistency. Relationships of $\delta^{18}$O$_{Cell}$ values between the two sites are still significantly positive 60 years after the fire impact. Dendroclimatic analysis indicates significant changes in tree-ring growth and isotopic ratio responses to climate due to the increased demand of water for trees during the post-fire period (deeper seasonal subsidence of permafrost).

**Keywords:** stable isotopes; tree-ring width; Siberia; vegetation cover; active layer thickness; wildfire impact; climatic response





## 1. Introduction

Approximately 20% of the boreal forests in the world are dominated by deciduous larch forests, mostly covering the areas with permafrost soils [1].

Globally, permafrost temperature for the 2007–2016 period has increased by $0.29 \pm 0.12$ °C [2], as a result of the increasing air temperature of almost 2 °C in the Northern Hemisphere over the past century [3]. Degrading permafrost can alter ecosystems, release large quantities of carbon dioxide ($CO_2$) and methane ($CH_4$) that influence the global climate, and thus the permafrost carbon feedback results in amplifying global climate warming and the resultant surface warming [4]. Recent observations in Russia and other northern ecosystems (Alaska, Canada and Scandinavia) show substantial warming of permafrost soils during the last 20 to 30 years, which continues into the 21st century [5–9]. Models confirm recent, and predict future decreases in permafrost stability from 1960 to 2100 along the Northern permafrost belt, beyond Siberia [10]. These changes in permafrost are an important factor, impacting tree growth and tree ecophysiology, which will be reflected in growth and stable carbon and oxygen isotope dynamics. In addition to climatic influences periodic wildfires need to be considered, which are common for the permafrost

zone and in turn affect the seasonal thawing of the soil for many years after such fire events, which significantly influence permafrost depth. In Northern Siberia the main limiting factors for growth in permafrost conditions are low air and soil temperatures, as well as shallow active soil layer (ASL) thickness [11–18]. The latter parameter varies considerably depending on the topography of the territory, aspect (north or south exposition), type of vegetation, and certainly on the frequency of wildfire events. The depth of active soil layer during the vegetation period may vary for different site conditions from 0.2 m to 2.0 m and more and typically reaches its maximum in September [15,16,19].

Periodic wildfires cause an increase in the depth of seasonal soil layer thawing, which persists for decades after the fire impact [20]. The frequency of forest fires for different regions with continuous permafrost in Siberia increases from north to south [21]. The fire return intervals in the forest-tundra ecotone can be as long as 275 years, while in the northern Taiga it is commonly only about 80 years [22,23]. For the study region (Evenkia) in the northern Taiga at 64° latitude the interval between fires varies from 27 to 200 years. Habitat conditions of burned territories can be changed significantly depending on the intensity of the fire, e.g., by reduced ground vegetation resulting in decreased albedo [15]. These changes have a significant impact on the dynamics of tree-ring growth, which is generally reflected in increasing annual growth increments of the surviving trees [1,23–26].

The variation of stable isotopes provides reliable information on environmental conditions, which is stored in tree-ring cellulose of different tree species and is characteristic for specific climatic impacts [27–30]. Consequently, in numerous studies, stable isotopes are used as a proxy for precipitation and temperature reconstructions [30–34]. The physiological aspects of concurrent carbon and oxygen discrimination in the metabolism of trees were applied to infer changes in water-use efficiency, photosynthesis or stomatal conductance [35–37]. Such data combinations may provide a better understanding of growth responses after ecosystem changes occurred, such as wildfires. Some studies of $\delta^{13}C$ and $\delta^{18}O$ in tree rings were focused on the permafrost zone of Eurasia [14,35,38–40]. However, for northern ecosystems, which are the most sensitive regions to climate change with dramatically changing conditions, only few studies so far have recorded a wide set of tree-ring parameters (TRW, maximum latewood density, cell structure and stable isotopes compositions) [14,41]. The influence of wildfires on all of these parameters has not yet been systematically investigated [42–45].

In this study we analyze TRW, $\delta^{13}C$ and $\delta^{18}O$ values in Gmelin larch tree-ring cellulose chronologies from Northern Siberia and addressed the following questions: (i) what are the main factors determining TRW dynamics and variations in stable tree-ring isotopes ($\delta^{13}C_{Cell}$, $\delta^{18}O_{Cell}$) in the permafrost zone of Siberia over the last 150 years; (ii) how do fires impact variations in $\delta^{13}C_{Cell}$, $\delta^{18}O_{Cell}$ and TRW.

## 2. Materials and Methods

### 2.1. Site Description

The study was conducted in the permafrost zone in the Evenkia region in the north of Central Siberia (Figure 1a). The climate is characterized as highly continental. According to the data from the meteorological station in nearby Tura (64°17′ N; 100°07′ E; 188 m a.s.l.) the mean annual temperature is −8.8 ± 1.4 °C for the 1936–2010 period, which increases for the observed period, from the mean value −9.4 ± 1.4 °C before 1980s to −8.3 ± 1.2 °C afterwards. The mean monthly air temperature of the warmest month (July) is 16.7 ± 1.6 °C (in some years the absolute day maximum was 35.5 °C). January is the coldest month with a mean temperature of −35.8 ± 5.0 °C (in some years the absolute day minimum was −67.1 °C in February [46]. The mean annual amount of precipitation is 369.0 ± 71.2 mm. During the vegetation season (June–August) about 47% of the annual precipitation is observed (Figure 1b). The precipitation sum is slightly increasing for the observation period, but not significant.

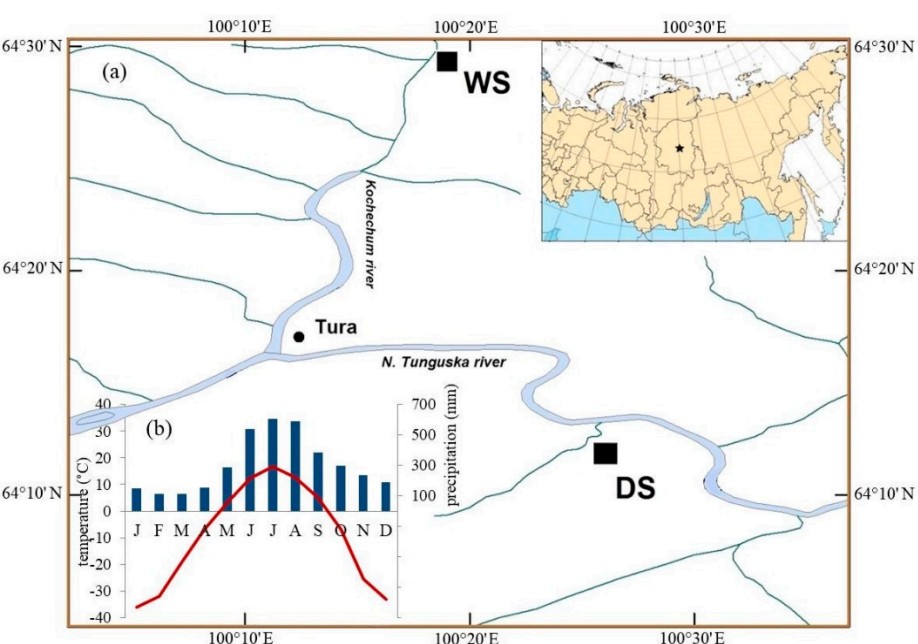

**Figure 1.** Map of the study sites (WS—wet site; DS—dry site) (**a**) and diagram with the main climatic characteristics of the study region Tura (**b**) (average values of precipitation (columns) and monthly mean temperature (line)) from the meteorological station Tura (1936–2010).

The studied area is located within the northern taiga. The main conifer tree species growing in the permafrost zone is Gmelin larch (*Larix gmelinii* (Rupr.) Rupr.). This species has a very wide range for survival with high thresholds to extreme temperatures [46].

The two compared sites differ considerably in their growth condition (Figure 1a). WS was described in [35] in the context of climate influence on stable isotope dynamics in larch trees under permafrost condition. Data for the DS were also partly presented in [42]. The studied tree stands grow on flat terrains that developed on deep fluvial deposits (second and third river terraces). The soil under such conditions is the Typic Haploturbel [47] with a low-thickness of the soil organic layer (10–20 cm). The depth of the ASL, defined as the portion of the upper soil that thaws in summer, reaches a maximum in September, between 0.2–0.9 m depending on the micro topography (mounds and troughs) induced by cryoturbation. Rooting depth in troughs is limited by the organic layer when the upper mineral soil is over moistened ($\geq$100% $v/v$) and cold (temperature in May at 10–40 cm is <1 °C). Rooting depths in mounds reach up to 40 cm with 80% of roots inhabiting the organic layer and upper 10 cm of mineral soil, which turns dry in July (20% $v/v$).

The two studied sites have contrasting ecological conditions with regard to topographic characteristics (high and low river terrace), forest stand characteristics (i.e., stand density, tree height, and diameters), ground vegetation (moss or lichen species as dominants in ground vegetation cover), thickness of the active soil layer and different fire history (number and years of fire, see Table 1). The abbreviated names for these two sites are WS (site under wet conditions with feather mosses dominating the ground cover at the high river terrace) and DS (site under dry conditions with lichen and dwarf shrubs dominating the ground cover at the low river terrace). For northern ecosystems in the Putorana mountains (Russia) [48] it was shown that the powerful development of lichen and mixed green moss-lichen cover is an indicator of dryness and nutrient poor soils.

**Table 1.** Site characteristics (statistical values are given with standard deviation).

| Parameters | Sites | |
|---|---|---|
| | **WS** | **DS** |
| Location | 64° 32′ N 100°14′ E<br>204 m a.s.l | 64°12′ N 100°27′ E<br>218 m a.s.l |
| Type of vegetation | larch with dwarf-shrubs, lichens (10%)<br>and mosses (90%) cover | larch with shrubs, dwarf-shrubs,<br>mosses (40%) and lichens (20%) cover |
| Depth of ASL, cm | 25.5 ± 21.9 | 57.7 ± 18.6 |
| Last detected fire events, year | 1852 | 1896 |
| Mean height of trees (H), m | 10.5 ± 6.2 | 16.0 ± 1.3 |
| Mean diameter on the breast height (DBH), cm | 8.7 ± 3.8 | 25.9 ± 3.2 |
| Forest density, N/ha | 1325 | 400 |
| TRW chronology, period | 1742–2005 | 1823–2011 |
| Mean TRW values, mm | 0.34 ± 0.19 | 0.64 ± 0.45 |
| Stable isotope chronology, period | 1864–2005 | 1860–2011 |
| Mean $\delta^{13}C_{Cell}$ values, ‰ VPDB | −22.93 ± 0.44 | −23.32 ± 0.57 |
| Mean $\delta^{18}O_{Cell}$ values, ‰ VSMOW | 23.40 ± 0.77 | 23.78 ± 1.0 |

The overstory vegetation at WS is represented by larch (*Larix Gmelinii*) with dwarf-shrubs (*Ledum palustre* L., *Vaccinium vitis-idaea* L.), lichens (10% area) and mosses (90%) in the ground cover (Table 1). For a detailed description see Sidorova et al. [35]. At the DS site the overstory vegetation is also represented by larch (*Larix Gmelinii*), and undergrowth with ground birch, dwarf birch (*Betula rotundifolia*, *B. nana*) and shrubby alder (*Duschekia fruticosa*). Ground vegetation of the DS site is represented by dwarf shrubs *Vaccinium uliginosum* and *V. vitis-idaea*, lichens (several species of *Cladonia* and *Cetraria* genera) and a few patches of feathermosses (*Pleurozium schreberi*, *Hylocomium splendens* etc.) (Table 1). The main contrasts of two sites are different seasonal thawing layer of soil (2/3 deeper in DS than WS) and type of soil cover vegetation (hypnum hydrophilic mosses are dominant at WS; lichens and draft-shrubs are dominant DS).

Tree-ring cores (one per tree) were sampled with an increment borer (0.5 cm diameter) at breast height from >30 larch trees of different ages. Tree-ring width (TRW) of all trees were measured with a precision of 0.01 mm (LINTAB V-3.0; Germany) and cross-dated for determining the exact calendar year for each tree-ring. The individual TRW series were standardized to remove non-climatic age trends in the raw data as follows: A cubic smoothing spline with 50% frequency-response cut off equal to 2/3 of the series length was fitted to the individual records and the residual tree-ring indices (as dimensionless indices) were calculated in ARSTAN [49]. Common signal strength of the individual tree chronologies (coherence of year-to-year variations of the tree-ring parameters) was estimated by inter-series correlation (rbar) and the expressed population signal (EPS) [50]. These statistics were calculated as an average of values obtained for a number of 50-year periods lagged by 25 years. Further, for the analysis, we used index chronologies obtained by averaging individual.

For the lichen dominated ground cover vegetation site, DS, the date with a fire event was identified for 1896 (the last date of fire in WS was determined to be 1852). To identify the date of fires, we used cross-sections of living trees (>3 for each sites) with after fire scars. A few cross-sections from larch trees at both sites were cross-dated with the master chronologies obtained from the study sites.

For the isotopic analyses, we selected five trees and took one or two cores depending on the mass of the annual ring at every year for each site. To eliminate the possible juvenile effect in isotope data, only material from trees older than 200 years at DBH was taken. The individual cores were split in a one-year resolution. Material for the same year from all of the cores was pooled [51,52] in accordance with the weight contribution of each sample and milled. Cellulose extraction was arried out according to [53]. Each sample was enclosed in Teflon bags and submersed in solutions of 5% NaOH and subsequently in 7% NaClO$_2$

(sodium chlorite). The samples were then washed in distilled water and dried at 50 °C. The isotope values for carbon ($\delta^{13}C$) and oxygen ($\delta^{18}O$) were determined on extracted $\alpha$-cellulose. For the determination of the C-isotope ratio, 0.6–0.8 mg were weighted into tin capsules and for the oxygen 1.1–1.3 mg were weighed into silver capsules. The $^{13}C/^{12}C$ and $^{18}O/^{16}O$ isotope ratios were determined separately by combustion or thermal pyrolysis [54], respectively, with two different Elemental Analyzers (EA1108, Finnigan, Germany and EA-1110; Carlo Erba, Milano, Italy, respectively) coupled to a mass spectrometer (Delta S, Finnigan) at the Paul Scherrer Institute (Switzerland). The long term, repeated analysis of standard materials (organics and water) yielded a standard deviation 0.1‰ for carbon and <0.2‰ for oxygen. The isotope values are expressed in delta notation relative to the international V-PDB reference for carbon and V-SMOW for oxygen. The methods used in this study for isotope analysis are described in detail in [55,56]. The obtained carbon isotope data were corrected for the decreasing $\delta^{13}C$ value of the atmospheric $CO_2$, which is a result of the release of $^{13}C$ depleted $CO_2$, originating from fossil fuel combustion and biomass burning in the course of land-use change. This correction was made with data from high precision records of atmospheric $\delta^{13}C$ in Antarctic ice cores, complemented by atmospheric measurements [57]. To keep information about physiological responses to "post-fire" effect we do not use a 'pre-industrial' ('pin') correction as proposed by McCarroll et al. [58], which is a constrained non-linear de-trending of the $\delta^{13}C$ series after AD 1850. The constraints are based on the potential physiological response of trees to increased $CO_2$.

To determine the influence of climate conditions on tree-ring parameters, correlation analyses between the chronologies and the instrumental records from Tura meteorological station were carried out.

### 2.2. Basic Principles of Isotope Fractionation in C3 Plants

For the data interpretation the basic principles between gas exchange and isotope fractionation according to Farquhar et al. [59], Farquhar and Loyd [60] and Scheidegger et al. [36] were applied. A consistent increase in $\delta^{13}C$ of tree-ring cellulose ($\delta^{13}C_{Cell}$) is either due to an increase of photosynthesis ($A_N$) at a given stomatal conductance ($g_l$), which is mostly an indicator for improved growth conditions concurrent with an increase in TRW (moist and warm climate under non limiting nutrient supply). Or such an increase in $\delta^{13}C_{Cell}$ can also occur as a result of decreasing $g_l$ at a given $A_N$, which is mostly an indicator for a decreasing water supply, increasing vapor pressure deficit (VPD) or drought. A decrease in $\delta^{13}C_{Cell}$ indicates either a decrease in $A_N$ at a given $g_l$ (low temperatures, decrease in nutrients or irradiation) or an increase in $g_l$ at a given $A_N$ (non-limiting water supply and low VPD often under cold climatic conditions). With regard to the oxygen isotopes an increase in $\delta^{18}O_{Cell}$ values occurs, with a decrease in $g_l$ and vice versa. Furthermore, an increase in $\delta^{18}O_{Cell}$ values is also determined by the $\delta^{18}O_{Source}$ of the source water taken up via roots (warm conditions and increasing VPD, e.g., warm summer rains) or with a reduction of $g_l$ [60,61]. A decrease in $\delta^{18}O_{Cell}$ is found with a decrease in the $\delta^{18}O$ of the source water, (colder conditions, or augmented use of permafrost water) or increased $g_l$ (non limiting water supply, high ambient humidity). The inverse relationship between $\delta^{18}O$ and $g_l$ is illustrated as follows: With e.g., increasing VPD the transpiration rate would increase as well if it were not for the stomatal closure counteracting an excessive water loss. As a consequence, $\delta^{18}O$ of the bulk leaf water increases since (i) the lighter $H_2^{16}O$ evaporates more readily as the heavier $H_2^{18}O$, leaving the bulk leaf water more enriched in $H_2^{18}O$ [62]; (ii) due to a diminished replenishment of the transpiratory water loss with unenriched xylem water (dilution of the $H_2^{18}O$ concentration of the bulk leaf water, described as Peclet effect [60,61]); and (iii) the reduced back diffusion of depleted atmospheric water vapor into the substomatal cavities [63].

For the calculation of a Running Correlation of the Isotope Dynamics (RCID) we applied the conceptual model, which is based on the dual C and O isotope approach [36], linking the measured $\delta^{13}C_{Cell}$ and $\delta^{18}O_{Cell}$ isotopes. Both, C and O isotope ratios are modified during $CO_2$ and $H_2O$ gas exchange in the same plant sample (i.e., tree-ring)

but by different processes (photosynthesis and transpiration). This procedure allows the distinction whether changes in the isotopic ratio is the result of a change in $A_N$ and/or $g_l$ as a response to environmental changes. Two situations (A and B) for the combined data set are compared representing the isotope and gas exchange results. A and B may represent two different points of time, or two different treatments in a controlled experiment or functional groups in an ecosystem (confers vs. broad leaf trees) etc. Based on the principles of gas exchange and isotopic fractionation mechanisms we can derive that an increase/decrease in $\delta^{13}C_{Cell}$ results either from an increase/decrease in $A_N$ at constant $g_l$ or a decrease/increase in $g_l$ at constant $A_N$. At the same time an increase/decrease in $g_l$ result in an inverse response (decrease/increase, respectively) in $\delta^{18}O_{Cell}$ [61]. While $\delta^{13}C_{Cell}$ varies due to changes of $A_N$ and $g_l$, $\delta^{18}O_{Cell}$ is only modified by the leaf as a function of $g_l$ only. Therefore, the combination of C and O isotope values from the same samples allows for the determination whether changes in stomatal conductance or/and photosynthesis responded to environmental changes [36].

When calculating a (RCID) between $\delta^{13}C$ and $\delta^{18}O$ over time, a positive correlation is found when both isotopes either increase or decrease concurrently. Negative correlations are found when either $\delta^{13}C$ increases, while $\delta^{18}O$ decreases or vice versa. Accordingly, the physiological interpretations can then be derived from the above outlined basic principles.

The data were statistically analyzed with the program package "Basic Statistics Version 6.0", applying the following routines: log-linear analysis, time series analysis, and multiple regression methods.

## 3. Results

### 3.1. Tree-Ring width, $\delta^{13}C$ and $\delta^{18}O$ Dynamics

Two TRW chronologies were analyzed for two sites, which cover the years 1823–2011 at the DS and the years from 1742 to 2005 at the wet moss-dominated site, WS. After the wildfires, a decrease of TRW with values below average for all after-fire periods ($0.36 \pm 0.22$ mm on DS and $0.33 \pm 0.20$ mm on WS) was observed during the first 8 at the DS and 18 years at WS, and TRW with values above average up to the 48 and 59 years after the date of fires, respectively (Figure 2a,b). Indices of TRW as well as raw data presented in relative units from the first years after the fire showed different length of dynamics with positive and negative values, but in general we observed increased growth for ±30 and ±50 years after the fire on DS and WS, respectively, followed by a decreasing growth.

The mean TRW values for the common period 1823–2005 are $0.34 \pm 0.19$ mm and $0.64 \pm 0.45$ mm for WS and DS, respectively (Table 1). Both series are characterized by a high inter-series correlation (rbar = 0.59 and 0.48, respectively) and a strong common signal (EPS = 0.95 and 0.91, respectively) and allow the development of representative population site records.

The standardized tree-ring index chronologies from WS and DS show a high inter-annual variability, and display a positive dynamic with increasing annual tree ring increments (post fire effect) already 8–18 years after the fire during the common period 1823–2005 (Figure 3a). However, the reaction to the fire at both sites is not the same as shown in Figure 2. The increase in TRW indices in WS after the fire in 1852 is more intensive and persistent, and is later followed by a significant negative trend. At DS, TRW index dynamics show a continual increase of TRW after the fire in 1896. Later, this trend is not so pronounced.

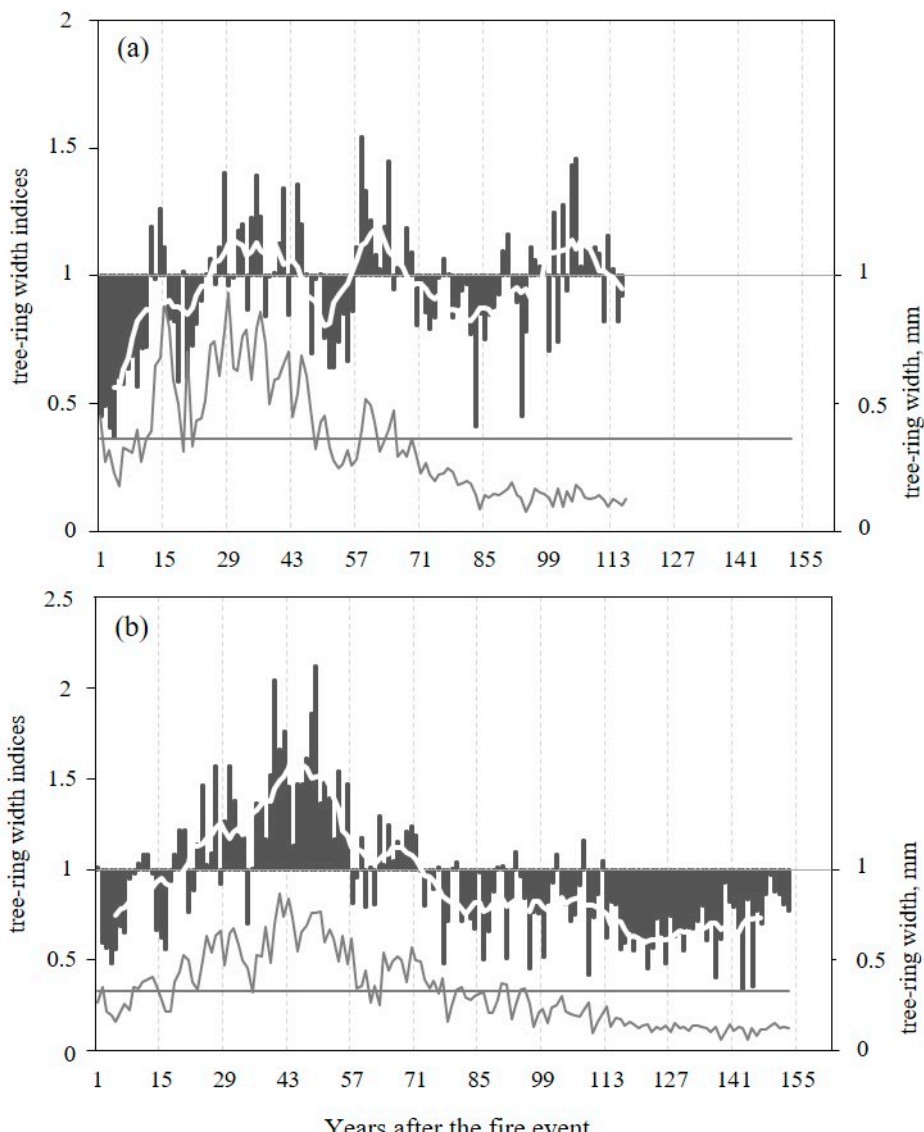

**Figure 2.** Dynamics of tree-ring width (line) and tree-ring width indices (column) for two sites from data (between 1896 and 2011 and between 1852 and 2005) after the detected fires DS—1896 (**a**) and WS—1852 (**b**), respectively, are shown. The white lines (based on the smoothening by an 11-years running means) indicate the periods with indices values > 1. The horizontal lines show the level of average values of TRW after fires (0.36 mm on DS and 0.33 mm on WS).

There are two periods of decline of $\delta^{13}C$ values in the end of the 19th century (before fire 1896) and in the middle of the 1970s for DS compared to WS, where a strong shift of more negative isotopes values was observed after 1960 (Figure 3b). Mean values of the carbon isotope ratios differ not so substantially between the two sites, but they are higher ($-22.93 \pm 0.44$‰) for WS in comparison with more negative values ($-23.32 \pm 0.57$‰) for DS (Table 1) across the whole period. Also, the mean values of $\delta^{13}C$ are not substantially different ($-23.75 \pm 0.46$‰ and $-23.36 \pm 0.61$‰, respectively) before and after the fire for the same time periods (30 years) for DS. However, if the variance between minimum and maximum values is compared before and after the fire 1896 for the same period (1.86‰ for 30 years before the fire versus 2.55‰ for 30 years after the fire), there is a clear increase in variance observed after the fire.

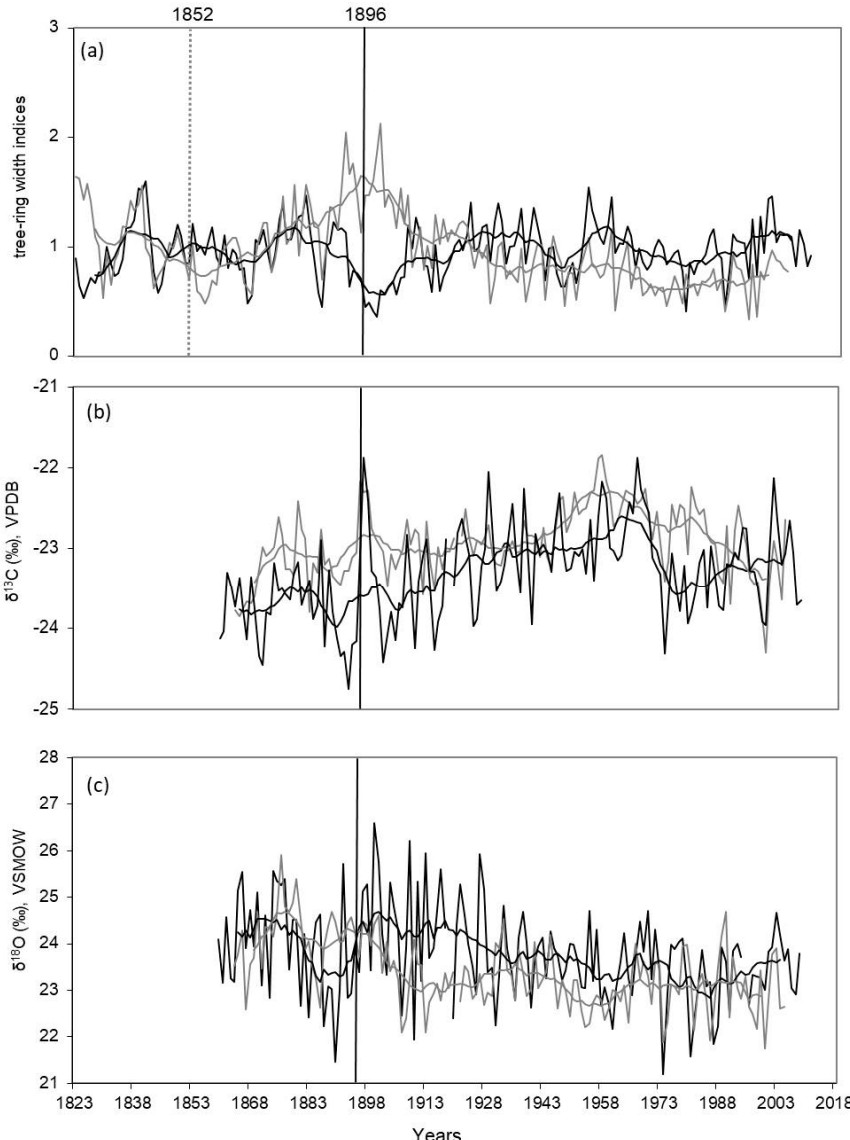

**Figure 3.** Dynamics of tree-ring width index (**a**), $\delta^{13}C_{Cell}$ (**b**) and $\delta^{18}O_{Cell}$ (**c**) chronologies for two comparative sites (black line, DS; grey line, WS) are shown. Smoothed data by a 11—years running means are shown. Vertical black line indicates the date of the fire in 1896 for DS, vertical dotted line indicates the date of the fire in 1852 for WS.

The dynamic of oxygen isotope ratios shows a common negative trend during the last 150 years for both DS and WS (Figure 3c). However, the $\delta^{18}O$ chronology for DS shows a strong reaction to a change in water supply from the fire effect in an $^{18}O$ enrichment for over 30 years after the fire. Mean values of $\delta^{18}O$ for the two sites are not significantly different ($23.40 \pm 0.77$‰ and $23.78 \pm 1.0$‰, for WS and DS, respectively) for the whole analyzed period. The $\delta^{18}O$ variance for the 30 years of the DS chronology before and after the fire differs not much (4.24‰ before and 4.65‰ after the fire event).

### 3.2. Relationships between Tree-Ring Parameters

To understand the character of changes in the dynamics of the studied parameters and their response to changes in ecological condition after the fire, we carried out statistical analyses with a running correlation applying a 30-years window. The correlations between DS and WS for TRW, $\delta^{13}C$ and $\delta^{18}O$ of larch tree-rings for the common period from 1864 to 2005 are shown in Figure 4. Furthermore, the calculated cross-correlations between these

parameters are presented in Table 2 for four equal time-blocks: before the fire in 1896 and in three periods of 30 years each after the fire.

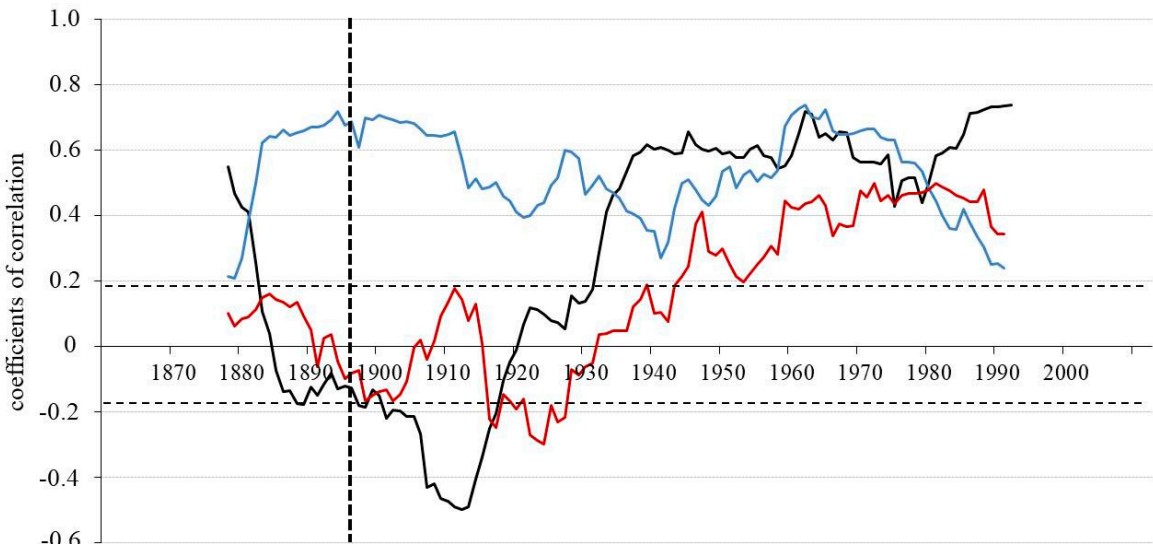

**Figure 4.** Running correlation between the two sites (WS and DS) for TRW (black line), $\delta^{13}$C (blue line) and $\delta^{18}$O (red line) chronologies applying a 30—years window (vertical dotted line indicates the fire event in 1896 at DS). The horizontal dotted lines indicate the threshold for significance at $p < 0.05$ (r = 0.19, $n$ = 113).

**Table 2.** Correlation coefficients (bold numbers are significant at $p < 0.05$) between all chronologies of DS and WS for the four selected periods before and after fire 1896 (30 years each).

| | | **WS** | | | | | | | | | | | |
| | | 1867–1896 | | | 1897–1926 | | | 1927–1956 | | | 1957–1986 | | |
| | | TRW | $\delta^{13}$C | $\delta^{18}$O | TRW | $\delta^{13}$C | $\delta^{18}$O | TRW | $\delta^{13}$C | $\delta^{18}$O | TRW | $\delta^{13}$C | $\delta^{18}$O |
|---|---|---|---|---|---|---|---|---|---|---|---|---|---|
| | TRW | **0.39** | **0.35** | **0.62** | **−0.46** | 0.03 | −0.36 | **0.65** | −0.27 | −0.08 | **0.49** | 0.16 | **−0.41** |
| DS | $\delta^{13}$C | **−0.38** | 0.18 | 0.05 | 0.09 | **0.66** | 0.12 | **−0.40** | 0.27 | −0.20 | 0.17 | **0.67** | 0.19 |
| | $\delta^{18}$O | −0.26 | 0.11 | 0.08 | 0.35 | 0.22 | 0.17 | 0.15 | 0.01 | 0.11 | 0.16 | 0.28 | **0.46** |

The standardized TRW chronologies show strong negative trends in running correlations till 1912, involving the period with fire effects 1896 in DS (Figure 4). After this time, a strong increase till 1930 is observed and then the relationship between TRW of the two sites became stable over time and remained constantly positive. For the whole period from 1860 to 2005 there was no significant correlation in TRW indices between WS and DS (r = −0.05; $p < 0.05$). However, the analysis for the 30-year period before the fire (1867–1896) shows a declining trend. Starting in 1880 it changed from a positive to a negative correlation between the two sites, while for $\delta^{13}$C it strongly increased. The relationship (r = −0.46; $p < 0.05$) between TRW indices for the two sites then remained negative in the first 30-year period after the fire. After 1926 it changed again to strong positive relationships (Figure 4 and Table 2).

Similar dynamics of running correlations are observed for oxygen isotope chronologies. Yet, the change from a negative to a positive relationship between the chronologies from the two sites, for both TRW and $\delta^{18}$O occurred at different times. Particularly, the reaction of TRW chronology was about 10 years earlier than that of the $\delta^{18}$O chronology. Correlation coefficients between $\delta^{13}$C for the two sites are highly positive for the complete time span (r = 0.57; $p < 0.05$), with the exception of the time before the fire and the second 30-years period after the fire, when the relationship of carbon isotopes is still positive (0.18 and 0.27), but not significant at $p < 0.05$.

The highest correlation between $\delta^{18}O$ for the two sites after 1957 (r = 0.46; $p < 0.05$) significantly contributes to a positive correlation for all three proxies for the whole time-period between 1860 and 2005 (r = 0.26; $p < 0.05$).

Since the studied sites burned at different times, we have carried out a running correlation analysis between $\delta^{13}C$ and $\delta^{18}O$ for each site separately to understand if there is a common factor, which influences the physiology of the trees and is thus reflected in the fractionation of carbon and oxygen isotopes under such extreme conditions (Figure 5). After a steady decrease in the correlation factors between 1870 and 1960, we observed a strong increase in the correlation between the two isotope ratios, and identified the "break point"~1960 for both sites, but more pronounced for WS, where such a point indicates the change of correlation between $\delta^{13}C_{cell}$ and $\delta^{18}O_{cell}$ from negative to positive. For DS this point indicates the changes from a period with a slightly decreasing dynamics of correlation between $\delta^{13}C$ and $\delta^{18}O$ to significantly increasing correlation.

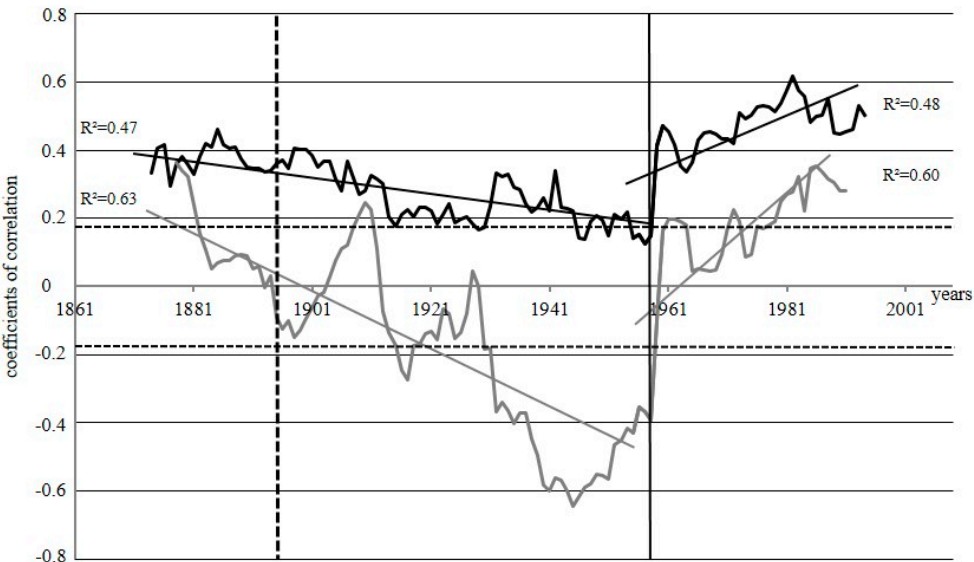

**Figure 5.** Running correlations between $\delta^{13}C$ and $\delta^{18}O$ by the 30—years window for DS (black line) and WS (grey line); the vertical line represents the "break point"~1960; the vertical dotted line indicates the fire event in 1896 at DS. The horizontal dotted lines indicate the threshold for significance at $p < 0.05$ (r = 0.19, n = 113). $R^2$—coefficient of regression of the approximation lines for different periods.

### 3.3. Climatic Response

To determine the influence of climatic changes on the dynamics on tree ring indices, $\delta^{13}C_{Cell}$ and $\delta^{18}O_{Cell}$ a common period for both sites was considered (1936–2005).

As was reported in Kirdyanov et al. [14] all tree-ring parameters (TRW, $\delta^{13}C_{Cell}$, $\delta^{18}O_{Cell}$) at the DS stand are positively correlated to summer temperatures. Positive correlations with mean of June-July-August (JJA) air temperatures are also confirmed in our analysis for all parameters (r = 0.26 for $\delta^{18}O_{Cell}$, r = 0.27 for $\delta^{13}C_{Cell}$ and r = 0.32 for TRW; $p < 0.05$). The highest correlation was found between $\delta^{13}C_{Cell}$ and July air temperature (r = 0.44; $p < 0.05$). The dynamic of the TRW and $\delta^{18}O_{Cell}$ is positively related with June temperature (r = 0.24 and 0.29, respectively, $p < 0.05$, Figure 6a).

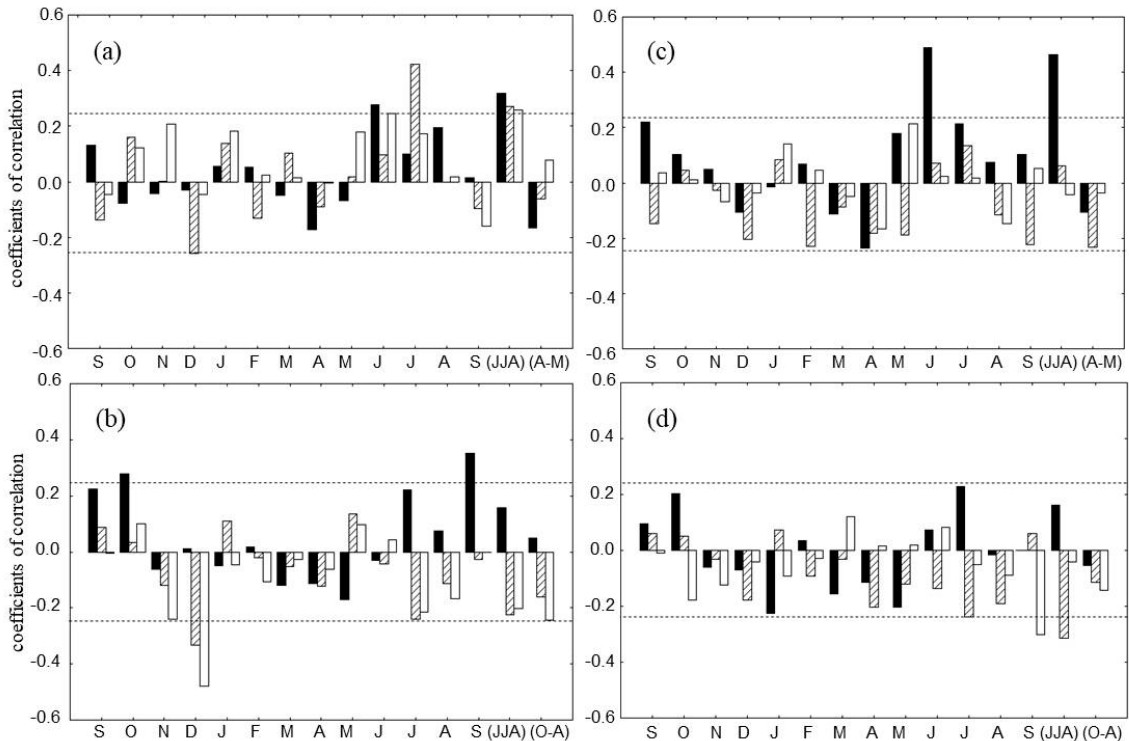

**Figure 6.** Tree responses to climate in DS (left panels) and WS (right panels) for the period 1936–2005. The relationships with climate are based on Pearson correlation coefficients between tree-ring data corresponding to each site and mean temperature (**a,c**) and total precipitation (**b,d**). Climate signals are investigated at monthly (previous September to current September) and seasonal time scales. For seasonal analysis we used mean of summer (June–August, JJA) and spring (April–May, AM) temperatures and sum of summer (JJA) and winter (October–April, O–A) precipitation. Tree-ring traits are represented by black (TRW chronology), hatched ($\delta^{13}C_{Cell}$), and white ($\delta^{18}O_{Cell}$) bars. The horizontal dotted lines indicate the threshold for significance at $p < 0.05$ (r = 0.24, n = 70).

The amount of October precipitation of the previous year and September precipitation of the current year are important for TRW (r = 0.27 and 0.32, respectively; $p < 0.05$) (Figure 6b). Carbon, as well as oxygen isotope ratios are negatively related with winter precipitation in particular for December (r = −0.33 and −0.48, accordingly; $p < 0.05$). July precipitation was negatively correlated with $\delta^{13}C_{Cell}$ as well as sum of winter precipitation with $\delta^{18}O_{Cell}$ (r = −0.24; $p < 0.05$).

The main factors limiting growth at WS are the temperature regime for the start of vegetation period and the July precipitation [28]. In our analysis the TRW chronology correlated with the summer temperature showing the strongest positive correlation in June and JJA (r = 0.49 and 0.46, accordingly; $p < 0.05$) (Figure 6c).

A negative relationship was obtained between $\delta^{13}C_{Cell}$ vs. sum of summer precipitations (r = −0.32; $p < 0.05$). September precipitation was negative related (r = −0.30; $p < 0.05$) to $\delta^{18}O_{Cell}$ for the whole observation time (Figure 6d).

Statistical relationships for all parameters with climate variables before (25 years) and after (25 years) of the "break point" (Figure 5) are not stable over time. Most important changes in climate response were found at the DS. The positive correlation with May precipitation was noted for carbon, as well as oxygen isotope ratios (r = 0.44 and 0.52, respectively; $p < 0.05$) and negative for TRW (r = −0.45; $p < 0.05$) before 1960. After 1960, July air temperature impacted isotopic variability significantly for d13Ccell (r = 0.57) and for d18O cell (r = 0.53) at $p < 0.05$, respectively

At the WS we observed negative relationship between temperature of May with $\delta^{13}C_{Cell}$ (r = −0.49; $p < 0.05$) and positive with $\delta^{18}O_{Cell}$ (r = 0.52; $p < 0.05$) before 1960, and positive relation between TRW and temperature of June (r = 0.52) after 1960. The most

stable relationships before and after "break point" were revealed between $\delta^{13}C_{Cell}$ and summer precipitation (r = −0.54 and −0.31 $p < 0.05$, accordingly).

The $\delta^{18}O_{Cell}$ values were significantly ($p < 0.05$) correlated with sum of summer precipitation (r = −0.45) and positive correlated with winter precipitation (r = 0.51, accordingly; $p < 0.05$) before 1960 only. The stable relationship in time relative to the 1960 between $\delta^{18}O_{Cell}$ and September precipitation (r = −0.51–0.55; $p < 0.05$) was found.

## 4. Discussion

Wildfires exert a strong impact on tree growth and the isotopic variability for both, WS and DS conditions. The main causes for fires in this region are summer thunderstorms under conditions such as prolonged summer drought events. These hot and dry phases occur periodically, resulting in forest wildfires in this region. Another factor contributing to a rapid spread of fires is the ground vegetation (green mosses and lichen), which can become exceptionally dry under high air temperature and low amount of summer precipitation [15]. With the incineration of the ground vegetation the natural thermal isolation layer is removed, enhancing the soil thermal flow, which leads to an increase of the permafrost depth and thus an increase of the ASL. The major long-term impacts of fires on surviving trees in boreal forest ecosystems are caused by changes of the soil thermal and hydrological regime and thus increase the availability of water and nutrients [64]. These factors represent a significant influence to all tree growth and physiological processes reflected in the variability of tree-ring parameters, such as TRW, $\delta^{13}C_{Cell}$ and $\delta^{18}O_{Cell}$. At the same time forest fires cause injuries on the cambial and root cells. A damaged root system impairs nutrient and water uptake, resulting in reduced photosynthesis and stomatal conductance reflected in $\delta^{13}C_{Cell}$ (immediate increase) and $\delta^{18}O_{Cell}$ (gradual and moderate increase), respectively (see discussion below) and diminished tree growth rates.

Thus, the initial decrease in TRW after the fire is most likely the result of a partially damaged tree root system, which is mainly located in the upper soil layers [46]. Furthermore, injuries in the cambium and meristem caused by the heat from the fire, take time to heal and recover. The subsequent positive trend in tree growth (until maximum of TRW) was found for up to ~30–50 years at DS and WS respectively (Figures 2 and 3a). The increase in radial growth after the fire is typical for both considered sites. The warmer soil climate improves the resource availability and root expansion, facilitating an increased tree growth. The absence or reduction of competition from other plant forms (i.e., dwarf shrubs, grasses, and mosses) also contribute to a meliorated resource availability, while the absence of the insulation layer consisting of dwarf shrubs, mosses and lichens sustains an enhanced soil thermal flow. Therefore, a positive trend in tree growth can be considered as a consequence of an enlarged ASL.

With the progression of recovery of the ground vegetation (in our case after ~30 and 50 years for DS and WS respectively) we observe a gradual decrease in tree growth (less pronounced at the dry site) as the thermal insulation of the soil increases, leading to a gradual reduction of the soil thermal flow and thus a recovery of the permafrost and reduced ASL [11,19]. That we can consider as a long-term ecosystem response.

While TRW declines after the fire for up to 8 and 18 years, the changes in stable C and O isotopes, reflecting tree physiological responses to post fire conditions showed up considerably faster response (short-term) [18,19]. In our study $\delta^{13}C_{Cell}$ in tree-rings from DS increases immediately after the fire. For WS we have no immediate post fire data; still the long-term trends look similar for both sites. We assume that the increase in $\delta^{13}C_{Cell}$ at WS in 1896 are presumably a response to either climatic impacts or direct fire damage of tree roots and needles (the increases in both isotope data and decrease in TRW indicate drought and higher temperatures, both preconditions for wildfires as occurred at DS). This increase in $\delta^{13}C$ results from a lower discrimination of the heavier $^{13}C$ isotope relative to the lighter $^{12}C$. High photosynthetic rates reduce the $CO_2$ concentration in the intercellular spaces in needles due to a higher $CO_2$ demand (increase in photosynthesis) or lower $CO_2$ supply (due to a reduced stomatal conductance) [57]. Here the immediate increase in $\delta^{13}C_{Cell}$ at

DS indicates an increase in $A_N$, rather than a reduction in stomatal conductance, which is reflected by a moderate increase of $\delta^{18}O_{Cell}$. As the isotopic ratios are formed in needles, the gas exchange performance on the needle level is expressed in the $\delta^{13}C_{Cell}$. Therefore, high $A_N$ rates on the single needle level are realistic, even though TRW values are low, the result of a much lower needle mass per tree after the immediate post fire period. Thus, the damaged tree crown produces only a limited amount of carbohydrates which is seen in a reduced TRW. The subsequent increase in TRW for the following decades (>30 to 50 years; see Figures 2 and 3) are due to the tree crown recovery, besides an improved water and nutrient availability. Consequently, trees can now fully exploit the enhanced resource pool from an enlarged ALS. The increase of TRW agrees well with increased $\delta^{13}C$ values.

The $\delta^{18}O_{cell}$ is given by the oxygen isotope ratio in the source water (precipitation and soil water) and leaf level $H_2^{18}O$ enrichment due to transpiration. Permafrost plays an important role in forming the oxygen isotope ratio as an additional source to water availability during the vegetation period. However, the available source water is strongly modified after the fire events due to: (i) the missing thermal insulation, which leads to an increased evaporation of soil surface water, causing an enrichment in $H_2^{18}O$ in the top soil, which can significantly influence $\delta^{18}O_{cell}$ in tree rings as a parameter of the ground-based evaporation and transpiration in the leaves [61]; and (ii) the summer precipitation, which is relatively enriched in $H_2^{18}O$ compared to permafrost water. This results in a composite of winter (low $\delta^{18}O$) and summer (higher $\delta^{18}O$) precipitation. Thus, the isotopic ratio of water taken up by the trees is, therefore, a very specific mixture of these two major water sources (precipitation and thawed permafrost water [18]). Furthermore, an increased active soil layer depth after fires can store more precipitation water, otherwise part of which would be lost as runoff water at a shallower permafrost depth. Finally, we conclude, that the fire effect in our study leads to drying of upper soil layers resulting in an increased use of the mix of precipitation and permafrost water in contrast to the absorption of mostly thawed permafrost water, which is predominantly used by the plants before the fire (Figure 4 in [11]). This results in an enrichment of $H_2^{18}O$ in the source water and ultimately in an increased $\delta^{18}O_{Cell}$ for the first five to eight years after the fire. After this short period, we observe a continuous decline in $\delta^{18}O_{Cell}$, which goes along with the recovery of the understory vegetation and thus a reduction of the soil heat flux, resulting in a decrease of the ASL and permafrost depth as well as a reduction in the surface evaporation, contributing to the decrease in $\delta^{18}O_{Cell}$.

The two sites show different climate responses, (see Figure 6), which is not surprising. The ecological conditions between the two sites clearly differ, i.e., soil properties, ground vegetation, tree traits etc. as shown in Table 2 (see sub-Chapter 2.1). Also, reaction of tree ring parameters to climate are differ at two sites over time, before and after of indicated "break point" according the relationships of running correlation dynamics between $\delta^{13}C$ and $\delta^{18}O$ (Figure 5). Since there are no significant changes in the long-term trends of individual climate variables (temperatures and precipitations of individual months), the fact of "break point" is a result various climatic response of tree ring parameters at different stages post-fire reforestation under various ecological condition of comparison sites (wet and dry).

To visualize the physiological responses after the fire event we performed a Running Correlation analysis between $\delta^{13}C_{Cell}$ and $\delta^{18}O_{Cell}$ Isotope Dynamics (RCID) for WS and DS (Figure 5) between 1870s and 2005. The remarkable decline of the correlation coefficients until 1960 for both sites reflects the diverging trend between the $\delta^{13}C_{Cell}$ and $\delta^{18}O_{Cell}$. As mentioned above the decrease in $\delta^{18}O_{Cell}$ indicates a decline of the soil temperature with an increasing scarcity of soil water resulting from a steady recovery to pre-fire conditions. The declining $\delta^{18}O_{Cell}$ is therefore a valid indicator for changes in the soil property indicating a reduction of the permafrost depth leading to a gradual resource limitation. In contrast the increase in $\delta^{13}C_{Cell}$ for ca. 30 years at DS, and for 50 years at WS (1852–1900) suggests an enhanced $A_N$ for as long as soil water and nutrients were sufficient to sustain an increased tree growth as reflected in TRW. Such trends with negative running correlations between

$\delta^{13}C_{Cell}$ and $\delta^{18}O_{Cell}$ at WS and DS can be explained by the gradual restoration of the ecosystem after fire event. It is well-known that after fires, ecosystems restore their species composition within approximately 50 years and return to the initial state (pre-fire) within 70–90 years [11,65,66]. The time for restoration, however, can vary considerably, depending on the species composition and soil properties, which will feedback on the heat flow into the soil altering the water balance of the ecosystem.

The remarkable change of RCID (from negative to positive) in 1960 results from a change to a unidirectional isotopic trend for both isotopes. After 1960 we find also a remarkable decline in $\delta^{13}C_{Cell}$ along with the slight but steady decline for $\delta^{18}O_{Cell}$. From that time on TRW was also remarkably low, indicating a significant reduction of $A_N$. However, the simultaneous and abrupt change at both sites in RCID in 1960 for all three tree-ring proxies indicates that other factors caused this "break point", i.e., changes in the microclimatic or hydrological regime (severe drought or intensive rains) and alterations in the nutrient availability. Besides we assume that a full restoration of the pre-fire soil cover vegetation with a shallower ASL limiting the water reserves and nutrient availability left the forest ecosystem more susceptible to stress factors resulting in such strong responses, as seen in 1960.

## 5. Conclusions

Irrespective of the dates of the fire events in Evenkia, our measurements showed that during the postfire period temperature, on the long term, was the dominant factor impacting TRW and stable C and O isotope ratios. The temperature impact became effective after the understory vegetation (mosses, shrubs and lichens) was incinerated by the forest fire, thus removing the thermic insulation. Although we considered two different dates of fire on two sites with different ecological conditions (wet and dry with different time span for recovery of understory vegetation as an isolating layer) the responses for the long-term post fire dynamics for all studied parameters are comparable and show similar patterns. As we have no data for the immediate post fire period at WS we cannot draw any comparative conclusions for this period. However, the site-specific characteristics and environmental impacts do modify the response velocity and intensity to the fire events and the subsequent tree response amplitudes, as reflected in tree-ring parameters (i.e., TRW, $\delta^{13}C_{Cell}$ and $\delta^{18}O_{Cell}$).

We determined that there are three stages of development after the wildfires Stage 1: After the fire we found a significant decline for TRW and an initial increase in the isotope values at DS ($\delta^{13}C_{Cell}$ and $\delta^{18}O_{Cell}$), followed by a rapid decline. The soil covering vegetation had been destroyed and fire injuries (needle buds, cambial cells and roots in the topsoil) impaired the tree's metabolic processes and growth, taking about 8 to 18 years for recovery; Stage 2: Without the cover vegetation thermal insulation was no longer effective causing the permafrost level to sink and the available water storage capacity to rise. This results in a thicker ASL along with its augmented nutrient and water availability, factors which foster growth and alter the isotope ratio (increase in $\delta^{13}C$ and decrease in $\delta^{18}O$) in tree rings. Under these beneficial conditions the ground covering vegetation and the ecosystem recovered over time to pre-fire conditions. With the gradual recovery of the understory vegetation (mosses, lichens and other species) the thermic insulation layer is restored, resulting in a decrease of the active layer depth; Stage 3 reflects the full reestablishment of the pre-fire conditions. It is characterized by a smaller permafrost depth and ASL and the fully recovered soil cover vegetation. A reduced TRW and lower $\delta^{13}C_{Cell}$ values suggest a decrease in carbon acquisition, due to a shallow ASL and reduced nutrient and water availability. It is noteworthy that this three-stage process is found in different sites, ecosystem types (dry and wet sites) and different dates of the wildfire event.

**Author Contributions:** A.V.K. collected samples; A.V.K. and A.S.P. contributed to fieldwork; A.A.K. and O.V.C. measured all presented data; A.A.K. and R.T.W.S. analyzed all data and prepared the manuscript. M.S. and R.T.W.S. supported the isotopes analyses. All authors have contributed equally and agreed to the published version of the manuscript.

**Funding:** This research was funded by the Swiss National Science Foundation Joint Research Project SCOPES (IZ73ZO_128035/1), SNF No. 200020_166162 (isotope measurements) and by the Russian Science Foundation (Project 18-14-00072-P).

**Data Availability Statement:** Data will be available on the open research data repository Zenodo upon the acceptance of the manuscript.

**Acknowledgments:** We thank the staff from the isotope laboratory at the Paul Scherrer Institute for their support with the sample preparation and analyses. The two anonymous reviewers are acknowledged for their valuable comments and suggestions, which contributed to the improvement of the manuscript.

**Conflicts of Interest:** The authors declare no conflict of interest.

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
