# Peer review of "Fire as a Major Factor in Dynamics of Tree-Growth and Stable δ13C and δ18O Variations in Larch in the Permafrost Zone"

_forests, doi:10.3390/f13050725_

Round 1
Reviewer 1 Report
The manuscript “Fire as a major factor in dynamics of tree-growth and stable 13C and 18O variations in larch in the permafrost zone” investigate the possible influences of forest fire on tree-ring growth and stable isotope chronologies as well as their relationship with climate factors. The study suggests that significant influences of forest fires on the responses of tree-ring growth and oxygen isotopic ratio to climate. The authors conclude that the δ13C chronologies are more consistent between the two sites and they are more adequate for climatic reconstruction. This manuscript is also well structured and written. The topic of this study is rather important and the results are interesting and the conclusions obtained have important implications. I only have a couple of minor comments/suggestions and hope they are helpful.
A main conclusion is that tree-ring carbon isotope chronology is a more adequate proxy for climate reconstruction. However, Figure 6 shows stronger correlations of tree-ring width and oxygen isotope chronologies with climate factors. Is the relationship of carbon isotope chronology with climate more stable over time? Nevertheless, I think more detailed analyses and discussion are needed to make this issue clear.
The author identified change break point at ~1960 CE between the two tree-ring isotope chronologies. I would like to know to what extent climate changes have contributed to this relationship change? To address this question, comparisons between the correlations of chronologies with climate for the periods before and after 1960 CE will be useful.
Author Response
Response to the Reviewer 1
Point 1: A main conclusion is that tree-ring carbon isotope chronology is a more adequate proxy for climate reconstruction. However, Figure 6 shows stronger correlations of tree-ring width and oxygen isotope chronologies with climate factors. Is the relationship of carbon isotope chronology with climate more stable over time? Nevertheless, I think more detailed analyses and discussion are needed to make this issue clear.
Response 1: Based on the Reviewers suggestion we performed additional detailed analyses (lines 377-392) and clarified text in the Discussion part (L. 478-486).
Point 2: The author identified change break point at ~1960 CE between the two tree-ring isotope chronologies. I would like to know to what extent climate changes have contributed to this relationship change? To address this question, comparisons between the correlations of chronologies with climate for the periods before and after 1960 CE will be useful.
Response 2: We performed additional analysis for all parameters (TRW, d13Ccell and d18Ocell) with climate data for two periods (before and after 1960) and clarified text in the part of Results (L.377-392) and Discussion (L. 477-484).

Reviewer 2 Report
Major revision
In this study, Knorre et al. explored tree-ring width (TRW) and stable isotope chronologies in tree-ring cellulose (δ13CCell, δ18OCell) from a wet (WS) and a dry (DS) site after the forest fires. They found TRW and δ18OCell were the most sensitive parameters in the changing tree growth conditions after fire. In contrast, the δ13CCell values in tree rings from the two sites are positively correlated independently of the fire impact. Finally, they concluded that 1) temperature, on the long term was the dominant factor impacting TRW and stable C and O isotope ratios during the postfire period; 2) the site-specific characteristics and environmental impacts do modify the response velocity and intensity to the fire events and the subsequent tree response amplitudes, as recorded by tree-ring parameters. The topic of fire and tree ring is refreshing and suitable for the readership of this journal. I recommend a major revision before this manuscript being accepted for publication.
Major comments:
- Lines 153-157, the fire years included 1896 and 1852. As shown by Swetnam 1996, fires are frequent in this region. Why only used these two fire years?
- How was the effect of fire on tree parameters (TRW, oxygen isotope, carbon isotopes) separated from climate? For example, in lines 394, the authors attributed the initial decrease in TRW to fires. However, the climate change also played a role. It is necessary to exclude the effect of climate change when discussing the effect of fires.
- Lines 14 or 502, wildfires and forest fires are used together which may cause confusion. To keep consistency, it is better to use one of them (wildfires or forest fires).
- Lines 346-354, it was found temperature was the major factor influencing tree-ring parameter. I would like to know the interannual comparisons of tree-ring parameters with temperatures.
References:
Swetnam, T. W. (1996). Fire and climate history in the central Yenisey region, Siberia. In Fire in ecosystems of boreal Eurasia (pp. 90-104). Springer, Dordrecht.
Author Response
Response to the Reviewer 2
Point 1: Lines 153-157, the fire years included 1896 and 1852. As shown by Swetnam 1996, fires are frequent in this region.
Response 1: Our study site is located towards northern part of Sieberia, while Swetnam 1996 studies western site (along the Dubches and Kac Rivers on the west side of the Yenisey River), where the frequency of wildfires less occurred according to Safronov and Volokitina (2010). Therefore, these two regions are remarkably different on the ecological conditions: there is continuous permafrost in the eastern side of the Yenisey River (our study area) and there is no one on the western side (study by Swetnam 1996). Permafrost availability and thawing permafrost under temperature increase impact significantly Siberian trees in our study site and this is confirmed in our study in 1960 – which is a break point recorded in our tree-ring proxies (TRW, d13Ccell, and d18Ocell).
Why only used these two fire years?
Response 1: For our study, we selected ecosystems with contrast ecological conditions where tree stands were survived after the fires and have longer tree longevity. In our case study we have dated only one fire event at each study site for common analyzing period (1823 -2005). We obtained a longer chronology at the WS compared to DS (1742-2005). We detected two fires else before 1852, but we did not discussed these fire events, because they lie outside of our common for two study sites analyzed period.
Point 2: How was the effect of fire on tree parameters (TRW, oxygen isotope, carbon isotopes) separated from climate? For example, in lines 394, the authors attributed the initial decrease in TRW to fires. However, the climate change also played a role. It is necessary to exclude the effect of climate change when discussing the effect of fires.
Response 2: Thank you for your comment. Fire signal can be preserved in tree rings for many years (long-term effect), opposite to short-term climate effect.
In general, the effect of decreasing growth and the reasons for this have been described repeatedly. The growth reduction can be seen as an abrupt narrowing of growth rings for several years after the fire (Stahlea et al. 1999; Schweingruber 1993). The potential impact of reduced water and nutrient supply, as well as photosynthate translocation due to an affected cambium, xylem, and phloem could potentially impact growth (Seifert et al., 2017). Also, Seifert et al. (2017) showed that post fire growth dynamics various significantly in different species and under different conditions.
In our study, a very important factor in the growth of trees and their response to a fire is highly dependent on the state of the seasonally thawed soil layer and, as a result, the availability of water (permafrost before the fire and atmospheric precipitation after the fire).
Of course you are right, the climate also plays very important role: for example, only during the hot and dry conditions of the growing season most fires occur. However, even with the rapid temperatures increase over the recent decades, we do not see such a sharp reaction of TRW to the climate as after the fires. Our results prove strong impact both factors, as fire as a climate, to tree-ring parameters and difficulties of divided the importance of one of them.
Point 3: Lines 14 or 502, wildfires and forest fires are used together which may cause confusion. To keep consistency, it is better to use one of them (wildfires or forest fires).
Response 3: In our text we used these two terms as synonyms, but we adjusted it to the Reviewer 2 requirements and left only wildfires. Thank you.
Point 4: Lines 346-354, it was found temperature was the major factor influencing tree-ring parameter. I would like to know the interannual comparisons of tree-ring parameters with temperatures.
Response 4: We did not performed seasonal TRW measurements. The annual comparison of tree-ring parameters with temperature shown in Figure 6. Additionally, we performed detailed climatic analysis for all studied tree-ring parameters (TRW, d13Ccell and d18Ocell) with climate data for two periods (before and after 1960) and clarified the text part of Results (L.377-392) and Discussion (L. 478-486).
References:
Schweingruber FH (1993) Trees and wood in dendrochronology. Springer Verlag Berlin Heidelberg, Berlin
Seifert, T., Meincken, M. & Odhiambo, B.O. The effect of surface fire on tree ring growth of Pinus radiata trees. Annals of Forest Science 74, 34 (2017). https://doi.org/10.1007/s13595-016-0608-8
Stahlea DW, Mushoveb PT, Cleavelanda MK, Roigc F, Haynesd GA (1999) Management implications of annual growth rings in Pterocarpus angolensis from Zimbabwe. For Ecol Manag 124:217–229
Sofronov, M.A., A.V. Volokitina. Wildfire ecology in continuous permafrost zone // Permafrost Ecosystems: Siberian Larch Forests, Ecological Studies. Еds. A. Osawa, O.A. Zyryanova, Y. Matsuura, T. Kajimoto, R.W. Wein. 2010. V. 209. P. 59–82.
